# The *Arabidopsis* J-Protein AtDjC5 Facilitates Thermotolerance Likely by Aiding in the ER Stress Response

**DOI:** 10.3390/ijms232113134

**Published:** 2022-10-28

**Authors:** Ting-Ting Shen, Lin Wang, Chun-Huan Shang, Yi-Cai Zhen, Yu-Lu Fang, Li-Li Wei, Ting Zhou, Jiao-Teng Bai, Bing Li

**Affiliations:** Ministry of Education Key Laboratory of Molecular and Cellular Biology, Hebei Collaboration Innovation Center for Cell Signaling and Environmental Adaptation, Hebei Key Laboratory of Molecular and Cellular Biology, College of Life Sciences, Hebei Normal University, Shijiazhuang 050024, China

**Keywords:** AtDjC5, thermotolerance, ER stress response

## Abstract

AtDjC5 belongs to the J-protein family in *Arabidopsis thaliana*. Its biological functions remain unclear. In this study, we examined the roles of AtDjC5 in resisting heat stress using reverse genetic analysis. After the seedlings were exposed directly to 44 °C for 90 min, *AtDjC5* knockout seedlings displayed decreases in the survival rate, membrane system stability, and cell vitality compared to WT seedlings, indicating that AtDjC5 is involved in plant basal thermotolerance. The *AtDjC5* knockout seedlings pre-exposed to 37 °C for 30 min exhibited decreases in the survival rate and total chlorophyll contents and increased cell death when they were subsequently exposed to 45 °C compared to the WT seedlings, indicating that AtDjC5 plays an important role in plant acquired thermotolerance. AtDjC5 was found to localize to the endoplasmic reticulum. The expression of the *AtDjC5* gene was induced by heat and TM (an ER stress inducer) treatment. Furthermore, we found that the knockout of *AtDjC5* inhibited ER stress-induced autophagy and the expression of ER stress-related genes. Taken together, these results suggest that AtDjC5 facilitates thermotolerance, likely by aiding in the ER stress response.

## 1. Introduction

A higher growing season temperature has great impacts on agricultural production, farm income, and food security [1,2,3]. High-temperature stress disturbs cellular protein homeostasis, leading to protein denaturation or aggregation [4,5,6]. Thermotolerance is an important component of organisms to adapt to high-temperature stress and is divided into basal and acquired thermotolerance [7,8]. Plants pre-exposed to a nonlethal high-temperature condition for a certain period of time exhibit an evidently increased tolerance to lethal high temperature. This is called acquired thermotolerance. The best-characterized aspect of acquired thermotolerance is the production of heat-shock proteins (HSPs), which are regulated by heat shock transcription factors [9,10,11,12]. HSPs function as molecular chaperones to allow cellular proteins to avoid and/or recover from stress-induced protein aggregation [13,14,15,16].

The accumulation of unfolded and/or misfolded proteins in the endoplasmic reticulum (ER) generates a condition called ER stress, which may activate the unfolded protein response (UPR), also called the ER stress response. The UPR is a stress mitigation mechanism that monitors protein folding in the cellular secretory pathway [17,18,19]. To alleviate ER stress, the UPR is activated to produce molecular chaperones or HSPs to aid in protein folding [20,21]. The molecular mechanism of the UPR in plant cells is much less well-understood than those in yeast and animal cells. In *Arabidopsis thaliana*, bZIP28 and bZIP60 of the basic domain/leucine zipper (bZIP) family have been identified as ER stress-activated membrane-related transcription factors [22,23]. bZIP60 mRNA is spliced by inositol-requiring protein 1 (IRE1) in response to ER stress, and the resulting mRNA encodes a nuclear targeted bZIP60, which activates the expression of UPR genes in the nucleus (IRE1-bZIP60 pathway) [24]. The activation of bZIP28 by unfolded proteins requires ER-localized molecular chaperones, BiPs (ER-localized HSP70s) [25]. Under nonstress conditions, BiPs bind to the lumen-facing domain of bZIP28. In response to ER stress, bZIP28 is released because BiPs bind to the unfolded protein accumulated in the ER, moves from the ER to the Golgi, and is then subjected to proteolysis by Golgi-resident proteases, which release its cytoplasmic domain, which relocates to the nucleus to activate the expression of UPR genes encoding chaperones, protein folding, and protein degradation factors such as BiPs (BiP-bZIP28 pathway) [23,24,26]. There are three BiP coding genes in *A. thaliana* (BiP1, BiP2, and BiP3), and BiP3, in particular, is the most highly upregulated by ER stress agents such as tunicamycin (TM) and dithiothreitol (DTT), two potent inducers of ER stress [27,28]. In addition, autophagy eliminates misfolded proteins by disposing of large protein aggregates and whole organelles through the formation of an autophagosome [29,30,31]. Plant autophagy has been shown to be involved in senescence, nutrient deprivation, oxidative stress, salt and drought stresses, and pathogen infection [32,33,34,35,36,37,38].

J-proteins (also known as Hsp40s) are defined by a J-domain consisting of approximately 75 conserved amino acid residues. They are located in various subcellular compartments and act as molecular chaperones, either alone or in association with HSP70s. Hsp70–J-protein chaperone machines play active roles in protein homeostasis by transiently binding to many different polypeptide substrates, restoring the native state of misfolded polypeptides trapped in aggregates [39,40]. One hundred and twenty J-proteins have been identified in the *Arabidopsis* genome [41,42]. J-proteins have been reported to regulate plant growth and development and to participate in the adaptation to various environmental stresses [43,44,45,46,47,48,49,50,51,52,53,54]. AtDjC5 is a member of the *Arabidopsis* J-protein family. The open reading frame of *AtDjC5* encodes a protein consisting of 113 amino acid residues. The roles of AtDjC5 in regulating the growth and development of plants and resistance to environmental stresses have not been reported. Based on our previous large-scale phenotype screening for the J-protein family with heat stress treatment, *AtDjC5* mutants possessed the heat sensitive phenotype. Therefore, in this work, we used reverse genetic analysis to define the roles of AtDjC5 in resisting heat stress in *A. thaliana.*

## 2. Results

### 2.1. Knockout of the AtDjC5 Gene Decreased Basal Thermotolerance

To understand the role of AtDjC5 in adapting to heat stress, we identified a homozygous T-DNA insertion mutant for the *AtDjC5 gene*, *djc5-1*. Sequence analysis of the T-DNA flanking regions revealed that *djc5-1* contained a T-DNA insertion in intron 2 of *AtDjC5* (Figure 1A). RT–PCR showed no detectable full *AtDjC5* transcript in the *djc5-1* plants (Figure 1B). To assess the effect of *AtDjC5* knockout on plant basal thermotolerance, the survival rates were compared between the WT and *djc5-1* plants after the seedlings were exposed directly to 44 °C for 90 min. The survival rate of the *djc5-1* seedlings was not different from that of the WT seedlings under normal growth conditions. However, the *djc5-1* seedlings exhibited hypersensitivity after heat shock (HS). While 49% of the WT seedlings were alive after HS, the survival rate after HS was 31% for the *djc5-1* seedlings (Figure 1C,D).

As an additional test of heat sensitivity, the electrolyte leakage level was compared between the WT and *djc5-1* leaves. Under non-HS conditions, there were no significant differences in the ion conductivity of the incubation medium between the WT and *djc5-1* leaves (Figure 1E). After HS at 45 °C for 120 min, the ion conductivity of the incubation medium was markedly increased in WT and *djc5-1*, and the ion conductivity in *djc5-1* was distinctly higher than that in WT (Figure 1E). The results indicated that AtDjC5 is important for the stability of the membrane system during HS.

To further confirm the role of AtDjC5 in plant thermotolerance, we compared the difference in cell vitality between the WT and *djc5-1* roots. Under non-HS conditions, the TTC reduction activity in the *djc5-1* roots was not obviously different from that in the WT roots; however, after HS at 45 °C for 1 h, the TTC reduction activity in the WT and *djc5-1* roots was markedly decreased, and the TTC reduction activity in *djc5-1* was markedly lower than that in WT (Figure 1F). HS prevented the reduction in TTC much more in *atdjc5-1* than in WT. Taken together, our data suggest that *AtDjC5* knockout accelerated HS-induced cell death and improved the basal thermotolerance of plants.

### 2.2. AtDjC5 Is Involved in the Regulation of Acquired Thermotolerance

To determine the role of AtDjC5 in plant acquired thermotolerance, we generated two other *AtDjC5* mutant lines (*djc5-2* and *djc5-3*) through the artificial microRNA method. RT–PCR showed no detectable full *AtDjC5* transcript in the *djc5-2* and *djc5-3* seedlings (Figure 2A). The 11-day-old seedlings grown at 22 °C were acclimated at 37 °C for 30 min and then returned to 22 °C for 120 min before challenge at 45 °C for 120 min. The survival rate of seedlings after HS was compared among WT, *djc5-1*, *djc5-2*, and *djc5-3*. The results showed that these three *AtDjC5* mutant seedlings were hypersensitive to HS. While 68% of WT seedlings were alive after HS, the survival rates after HS were 56% for *djc5-1*, 46% for *djc5-2*, and 51% for *djc5-3* (Figure 2B,C). The decrease in acquired thermotolerance in the *AtDjC5* mutants was confirmed by testing the total chlorophyll contents of seedlings. Before HS, the total chlorophyll contents in the three *AtDjC5* mutant lines were not obviously different from those in WT; however, the total chlorophyll contents in the three *AtDjC5* mutant lines were markedly lower than those in WT after HS (Figure 2D). As an additional test of heat sensitivity, heat-induced cell death was assayed in WT, *djc5-1*, *djc5-2*, and *djc5-3* by trypan blue staining. The three *AtDjC5* mutant lines exhibited increased cell death after HS compared to WT (Figure 2E).

Furthermore, we examined the effect of *AtDjC5* overexpression on plant acquired thermotolerance. A binary vector containing the *AtDjC5* coding region (*pCAMBIA1300-35S::AtDjC5*) or an empty vector (*pCAMBIA1300-35S*) was transformed into Col plants. PCR assays confirmed that the *AtDjC5* gene was introduced into all *AtDjC5* transgenic lines obtained. Q-PCR assays showed that the levels of *AtDjC5* mRNA were strongly increased in the OE1 and OE2 overexpression lines compared to the WT and transgenic lines containing an empty vector (EV) (Figure 3A). The WT, EV, and *AtDjC5* overexpression seedlings grown at 22 °C were acclimated at 37 °C for 30 min and then returned to 22 °C for 120 min before challenge at 45 °C for 120 min. While 50% of the WT seedlings were alive after HS, the survival rates after HS were 44% for EV, 66% for OE1, and 93% for OE2 (Figure 3B,C), indicating that *AtDjC5* overexpression increased the acquired thermotolerance. Taken together, our data suggest a role for AtDjC5 in improving the acquired thermotolerance of *A. thaliana* seedlings.

### 2.3. Expression Pattern and Subcellular Localization of AtDjC5

Bioinformatics analysis showed that AtDjC5 may exist in the cytosol, endoplasmic reticulum (ER), Golgi apparatus, and other cellular compartments (http://utoronto.ca/cell_efp/cgi-bin/cell_efp.cgi). We examined the subcellular localization of AtDjC5 using transgenic plants stably expressing 35S::AtDjC5-sGFP. ER-Tracker Red, an ER-specific fluorescent dye, was used to highlight the ER in the cells of root hairs. Merged images of GFP fluorescence (green) and ER-Tracker Red fluorescence (red) showed bright yellow fluorescence, indicating that the AtDjC5-sGFP fusion protein was partially present in the ER (Figure 4A).

The spatiotemporal expression pattern of the *AtDjC5* gene was examined using two approaches. First, the GUS reporter assay was used to monitor the activity of the promoter of the *AtDjC5* gene. The results obtained from transgenic plants harboring *PAtDjC5::GUS* showed that *GUS* under the *AtDjC5* promoter was expressed in cotyledons, roots, stems, cauline leaves, flowers, and young siliques (Figure 4B). Then, Q-PCR was used to compare the expression levels of *AtDjC5* in different tissues. *AtDjC5* was expressed to different degrees in all of the tissues analyzed, consistent with the results obtained by the GUS reporter assay (Figure 4C).

Furthermore, we investigated the effect of heat shock treatment on the expression of the *AtDjC5* gene using Q-PCR. Ten-day-old seedlings grown at 22 °C were exposed to 37 °C for 0, 15, 30, 60 or 90 min, which induced an increase in the expression level of the *AtDjC5* gene. The level of *AtDjC5* mRNA began to increase 30 min after HS and continued to rise until 90 min (Figure 5A). Due to the ER localization of *AtDjC5*, we evaluated the effect of tunicamycin (TM) treatment, an ER stress inducer, on *AtDjC5* gene expression. After the 10-day-old seedlings grown at 22 °C were treated with 5 μg/mL TM for 2 h, the expression of the *AtDjC5* gene was markedly increased, reaching 3.5 times that detected before treatment (Figure 5B). The data above suggest that the expression of the *AtDjC5* gene was induced by heat stress and ER stress.

### 2.4. AtDjC5 Is Required for ER Stress–Induced Autophagy and the ER Stress Response

To clarify whether AtDjC5 is related to ER stress-induced autophagy, the induction of autophagy was examined in WT and *djc5-1* mutant plants. Seven-day-old WT and *djc5-1* seedlings were treated with 5 μg/mL TM for 8 h, followed by MDC staining; nontreated seedlings were used as a control. Both WT and *djc5-1* seedlings showed very few autophagosomes in the control group (Figure 6A). The WT seedlings showed markedly elevated autophagy after TM treatment; however, autophagy induction was not observed in response to TM treatment in *djc5-1* seedlings (Figure 6A). To quantify these results, autophagosome numbers were analyzed per root section for WT and *djc5-1*. Autophagosome numbers in the WT roots strongly increased after TM treatment; however, the *djc5-1* roots did not show obvious differences from WT roots in autophagosome numbers after TM treatment (Figure 6B), indicating that AtDjC5 is likely to be involved in the regulation of ER stress-induced autophagy.

Furthermore, we explored the possibility of AtDjC5 participating in UPR regulation. We examined the effects of TM treatment on the expression levels of the *BiP1*, *BiP2*, *BiP3*, *bZIP17*, *bZIP28*, and *bZIP60* genes. Except for *bZIP17*, the expression levels of the other five genes were upregulated to different degrees in the WT and *djc5-1* seedlings after treatment with 5 μg/mL TM for 30 min (Figure 7A–F). Meanwhile, we found that *AtDjC5* knockout significantly suppressed the TM-induced upregulation of *BiP3* and *bZIP28* expression (Figure 7G). The above results indicate that AtDjC5 is required for the ER stress response.

## 3. Discussion

In plants, J-domain proteins have been reported to localize in different subcellular compartments and participate in various biological processes. A previous study showed that AtDjC5 (also named AtPam18-3) mainly exists in mitochondria [55]. However, in this study, we found that AtDjC5 showed a partial overlap with the ER maker (Figure 4A), indicating that AtDjC5 possibly has multiple subcellular localization.

Eukaryotic have two evolutionarily highly conserved systems to adapt to environmental stress conditions: the heat shock response (HSR) and the UPR. Various environmental stresses may lead to the accumulation of unfolded proteins or denatured proteins in plant cells. HSPs/chaperones respond to environmental stress, while cells respond to the accumulation of unfolded proteins in the ER by activating the UPR to generate HSPs/chaperones. It is likely that there is crosstalk between the HSR and the UPR. Heat stress has been shown to transiently induce X-box binding protein 1 (XBP1) splicing and to lead to increases in the mRNA expression levels of HSPA5 and DNAJB9 (ERdj4) [56,57,58,59], two typical UPR genes. The IRE1-bZIP60 pathway in plants is activated by heat stress [56]. Gao reported that bZIP28 is an important component of the plant response to heat stress and is involved in the regulation of chaperone expression [60]. Further work has confirmed the role of bZIP28 in the plant response to heat stress and connected it to the maintenance of fertility under heat stress [61]. In this study, we showed that AtDjC5 was induced by heat and TM treatment, and the knockout of *AtDjC5* led to decreased thermotolerance and inhibited the ER stress-induced upregulation of *BiP3* and *bZIP28* expression (Figure 5 and Figure 7), suggesting that AtDjC5 promotes plant thermotolerance, likely by assisting the BiP3-bZIP28 pathway. The dissociation of BiP3 from bZIP28 is a major switch that activates the UPR signaling pathway in plants. In response to heat stress, BiP3 competes with unfolded proteins in the ER to release bZIP28, allowing it to be mobilized from the ER to the Golgi apparatus and then enter the nucleus to activate the UPR. AtDjC5, as a chaperone, is likely to play an important role in assisting the dissociation of BiP3 from the C-terminal tail of bZIP28. Loss of AtDjC5 function might block bZIP28 exit from the ER and inhibit the UPR, which downregulates the expression of chaperone proteins, leading to decreased thermotolerance. Whether AtDjC5 plays a role as a partner of BiP3 and how they regulate UPR still need answers.

## 4. Materials and Methods

### 4.1. Plant Materials and Growth Conditions

Seeds of *Arabidopsis thaliana* (ecotype Columbia-0) were surface-sterilized, plated on Murashige and Skoog (MS) medium containing 1.0% (*w*/*v*) sucrose and 0.8% (*w*/*v*) agar, and kept at 4 °C for 3 d. Then, plants were cultured in a growth chamber under long-day conditions (16/8 h photoperiod) at approximately 100 μmol photons m^−2^ s^−1^ and 22 °C. Two-week-old seedlings were transplanted to soil and cultured under the original growth conditions.

### 4.2. Identification and Isolation of AtDjC5 T-DNA Insertional Mutants

Seeds of a putative T-DNA insertional mutant for *AtDjC5* (At5g03030) and SALK_091892 (*djc5-1*) were obtained from the Arabidopsis Biological Resource Center (Columbus, OH, USA). The homozygous *AtDjC5* mutant was identified by PCR as described by the Salk Institute Genomic Analysis Laboratory (http://signal.salk.edu/tdna_FAQs.html). PCR was conducted with genomic DNA from T2 generation seedlings using *AtDjC5*-specific primers (LP, 5′- GCGTCTGTAATCGACGGTAAG-3′; RP, 5′-CTGGCTATACACAGGCTACGC-3′) and a T-DNA left border primer LBb1.3 (5′-ATTTTGCCGATTTCGGAAC-3′). The position of the T-DNA insertion was determined by PCR product sequencing.

The transcript levels of *AtDjC5* in WT and *djc5-1* were detected by RT–PCR using a RT–PCR Kit (TaKaRa, Otsu, Japan). Total RNA was isolated from 10-day-old seedlings with TRIzol reagent (Life Technologies, Carlsbad, CA, USA). The *AtDjC5* coding region was amplified using a forward primer (FP) 5′- ATGGCTACGCCAATGATTGC-3′ and a reverse primer (RP) 5′-TCAAAAGGCAGAACCGCTGT-3′. *Actin* was used as a loading control and amplified using FP 5′-AGGCACCTCTTAACCCTAAAGC-3′ and RP 5′-GGACAACGGAATCTCTCAGC-3′. PCR products were analyzed by agarose gel electrophoresis.

### 4.3. Heat Stress Treatment and Chlorophyll Measurement

Seeds of different genotypes were planted on separate regions of the same MS plate, with 30 seeds per genotype and experiment. For the basal thermotolerance assay, 11-day-old seedlings grown at 22 °C were exposed to 44 °C for 90 min and recovered at 22 °C for 3 to 7 d prior to calculating the survival rate. For the acquired thermotolerance assay, 11-d-old seedlings grown at 22 °C were acclimated at 37 °C for 30 min and returned to 22 °C for 120 min before challenge at 45 °C for 120 min. Then, the seedlings recovered at 22 °C for 3 to 7 d prior to calculating the survival rate or analyzing the total chlorophyll content. Plants that were still green and produced new leaves were scored as surviving. The total chlorophyll content was determined as described by Porra [62].

### 4.4. Electrolyte Leakage Measurement

Membrane system stability was evaluated by an electrolyte leakage assay before and after heat treatment. Green leaves from 11-day-old seedlings grown at 22 °C were washed three times with ion-free water and then incubated in 5 mL of ion-free water at 45 °C for 120 min. The conductivity of the incubation medium was measured using a Leici conductivity meter (DDS-IIA, Shanghai, China).

### 4.5. Assay of Root Vitality

The vitality of the root tissue was detected with 2,3,5-triphenyltetrazolium chloride (TTC) according to Gong et al. [63] with modifications. Eleven-day-old seedlings grown at 22 °C were exposed to 45 °C for 1 h. Heat-treated seedlings were transferred to 0.6% (*w*/*v*) TTC solution and cultured at 22 °C for 20 h. After washing three times, 1.0-cm tips of primary roots (total 30 per sample) were homogenized in 95% (*v*/*v*) ethanol and centrifuged at 4 000× *g* for 5 min. The absorbance of the supernatant at 530 nm was measured using a spectrophotometer (TU-1900, PERSEE, Beijing, China).

### 4.6. Plant Transformation

To generate the *P35S::AtDjC5* construct, the *AtDjC5* coding region was PCR-amplified with cDNA from *A. thaliana* (Columbia-0) seedlings using FP 5′-TCTAGAATGGCTACGCCAATGATT-3′ and RP′-GAGCTCTCAAAAGGCAGAACCGCT-3′. The PCR product was cloned into the binary vector *pCAMBIA1300-35S* digested with *Xba* I/*Sac* I.

To generate the *PAtDjC5::β-glucuronidase* (*GUS*) construct, a 1239-bp DNA fragment upstream of the *AtDjC5* translational start codon was PCR-amplified with genomic DNA from *A. thaliana* seedlings using FP 5′-CTGCAGAACTCCCTCAAGGCTAAACC-3′ and RP 5′-TCTAGATATTCAGCTAAGTAGTTGTTCGG-3′. The PCR fragment was ligated into the binary vector *pCAMBIA1300-GUS* digested with *Pst* I/*Xba* I.

To generate the *P35S::AtDjC5-sGFP* construct, the *AtDjC5* coding region was obtained from a plasmid containing the *AtDjC5* coding region by digestion with *Xba* I/*Sac* I. The *AtDjC5* coding region was fused to the N terminus of sGFP in the binary vector *pCAMBIA1300-35S::sGFP* digested with *Xba*I/*Sac*I.

The resulting plasmids were introduced into *Agrobacterium tumefaciens* (GV3101). Transformation of *A. thaliana* was performed by the floral-dipping method [64]. Transgenic lines were selected on MS medium containing 25 μg mL^−1^ hygromycin.

### 4.7. Real-Time Quantitative RT–PCR

Total RNA was isolated from different tissues or different genotypic seedlings using TRIzol reagent (Life Technologies, Carlsbad, CA, USA). Real-time quantitative RT–PCR (Q-PCR) was performed as described by Zhang et al. [65]. Primer pairs were designed using Primer Express software (Applied Biosystems) (Appendix A). *ACTIN* was used as the internal control.

### 4.8. Obtaining AtDjC5 amiRNA Mutants

Using WMD3-Web MicroRNA Designer (http://wmd3.weigelworld.org/cgi-bin/webapp.cgi, accessed on 10 February 2016), we obtained two *AtDjC5*-specific artificial microRNAs (amiRNAs) (“TAAGCAAGATAGTTACTGCCT” and “TTCTGGGTGATTTACAACCAT”) and four oligonucleotide sequences (I to IV) (Appendix A), which were used to engineer amiRNA into the endogenous miR319a precursor by site-directed mutagenesis. The detailed protocol for amiRNA cloning is as described by Schwab [66]. The resulting two amiRNAs were cloned into the binary vector *pCAMBIA1300* via the *BamH* I/*EcoR* I site. The resulting plasmids were introduced into *Agrobacterium tumefaciens* (GV3101). Transformation of *A. thaliana* was performed by the floral-dipping method [64]. Transgenic lines were selected on MS medium containing 25 μg mL^−1^ hygromycin. The transgenic plants introduced into the above two amiRNAs were defined as *djc5-2* and *djc5-3*, respectively.

### 4.9. Subcellular Localization of AtDjC5

Roots of 5-day-old transgenic seedlings harboring *P35S::AtDjC5-sGFP* were incubated in 1 μM ER-Tracker Red (Molecular Probes, Carlsbad, CA) solution for 30 min in the dark. Green fluorescent protein (GFP) fluorescence and ER-Tracker Red fluorescence were observed using a laser scanning confocal microscope (FV3000; OLYMPUS, Tokyo Prefecture, Japan) with excitation/emission wavelengths of 488/510 nm for GFP and 587/615 nm for ER-Tracker Red.

### 4.10. Histochemical Staining

Trypan blue staining was conducted as described by Choi et al. [67]. Histochemical staining for GUS expression was performed according to Jefferson et al. [68].

### 4.11. MDC Staining and Microscopy

Seven-day-old WT or *djc5-1* seedlings were incubated in MS liquid medium with or without 5 μg/mL TM for 8 h, subjected to MDC staining with 0.05 mM monodansylcadaverine (MDC) for 10 min, and washed three times with PBS. MDC fluorescence was visualized using a laser scanning confocal microscope (LSM 710; Zeiss, Oberkochen, Germany) with an excitation/emission of 335/508 nm.

### 4.12. Statistical Analysis

Statistical analyses were carried out with STATISTICA 6.0 (StatSoft, Inc., Tulsa, OK, USA). The significance of differences was determined at *p* < 0.05 using ANOVA with Tukey’s HSD test.

## Figures and Tables

**Figure 1 ijms-23-13134-f001:**
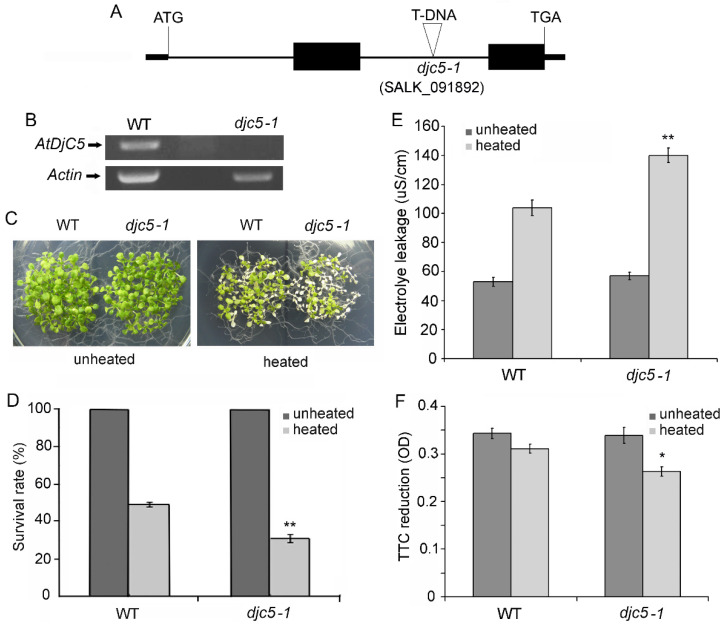
*AtDjC5* knockout decreased the basal thermotolerance. WT, wild-type plants; *djc5-1*, a T-DNA insertion mutant line for the *AtDjC5* gene; unheated, under normal conditions; heated, treated at 44 °C for 90 min. (**A**) Intron/exon organization of the *AtDjC5* coding region and T-DNA insertion location. Solid boxes, exons; lines, introns; triangles, T-DNA insertion position. (**B**) RT–PCR analysis of the *AtDjC5* full transcript in WT and *djc5-1* plants. (**C**) Comparison of WT and *djc5-1* seedling viability. (**D**) Survival rates of WT and *atdj5c-1* seedlings. Each value is the mean ± SD of ten biological replicates. (**E**) Electrolyte leakage assay for the WT and *djc5-1* leaves. The data are the means ± SDs of three independent experiments. (**F**) 2,3,5-triphenyltetrazolium chloride (TTC) reduction activity of roots for WT and *djc5-1*. The data are the means ± SDs of four biological replicates, with a total of 30 roots per sample. Asterisks indicate significant differences from the WT of the same treatment (*t* test, * *p* < 0.05, ** *p* < 0.01).

**Figure 2 ijms-23-13134-f002:**
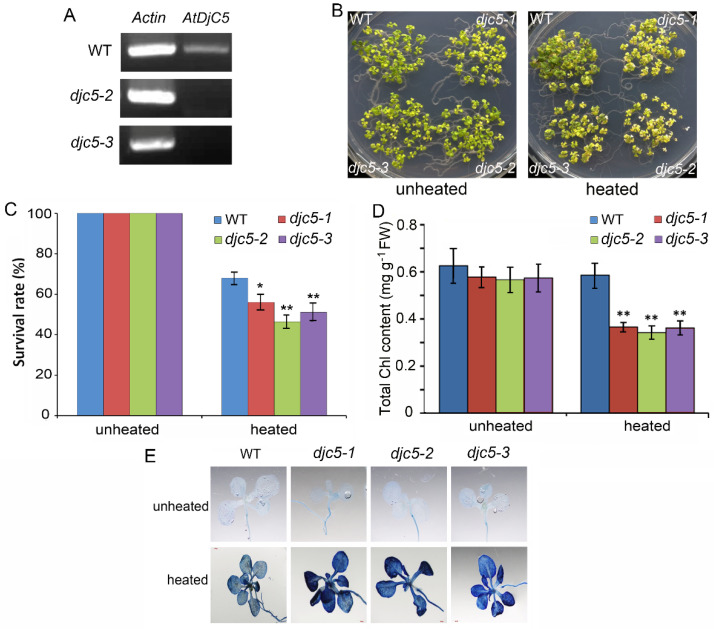
*AtDjC5* knockout decreased the acquired thermotolerance. WT, wild-type plants; *djc5-1*, a T-DNA insertion mutant line for *AtDjC5* gene; *djc5-2* and *djc5-3*, two *AtDjC5* mutant lines by artificial microRNA; unheated, under normal conditions; heated, treated at 37 °C for 30 min, then 22 °C for 120 min, followed by 45 °C for 120 min. (**A**) RT–PCR analysis of the *AtDjC5* full transcript in WT, *djc5-2,* and *djc5-3* plants. (**B**) Comparison of the WT and *AtDjC5* mutant seedling viability. (**C**) Survival rates of WT and *AtDjC5* mutant seedlings. Each value is the mean ± SD of ten biological replicates. (**D**) Total chlorophyll contents in WT, *djc5-1*, *djc5-2,* and *djc5-3* leaves. The data are the means ± SDs of four biological replicates. (**E**) Analysis of cell death in the WT and three *AtDjC5* mutant lines before and after heat stress by trypan blue staining. Asterisks indicate significant differences from the WT of the same treatment (*t* test, * *p* < 0.05, ** *p* < 0.01).

**Figure 3 ijms-23-13134-f003:**
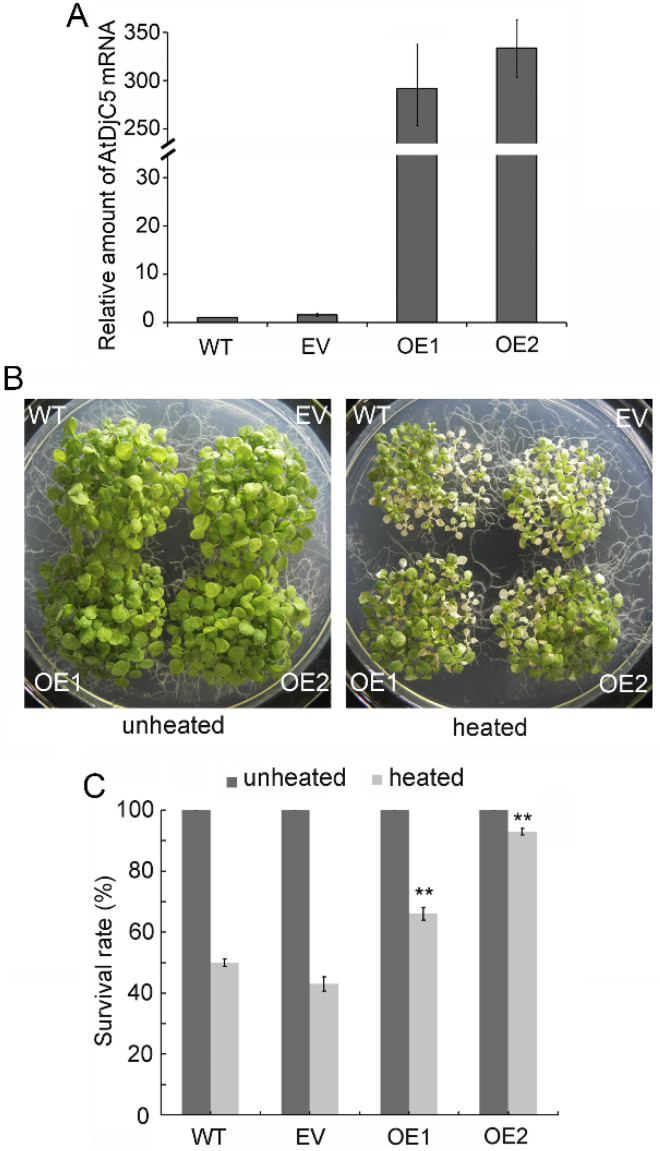
*AtDjC5* overexpression increased the acquired thermotolerance. WT, wild-type plants; EV, a transgenic line containing empty vector *pCAMBIA1300-35S*; OE1 and OE2, two *AtDjC5* overexpression lines. (**A**) Analysis of *AtDjC5* transcript levels through real-time quantitative RT–PCR (Q-PCR). The *AtDjC5* transcript level in the WT sample was used as the calibrator and was set to 1. Data are the means ± SDs of three biological replicates. (**B**) Comparison of WT and *AtDjC5*-overexpressing seedling viability. (**C**) Survival rates of WT and *AtDjC5* overexpression seedlings. Each value is the mean ± SD of ten biological replicates. Asterisks indicate significant differences from WT of the same treatment (*t* test, ** *p* < 0.01).

**Figure 4 ijms-23-13134-f004:**
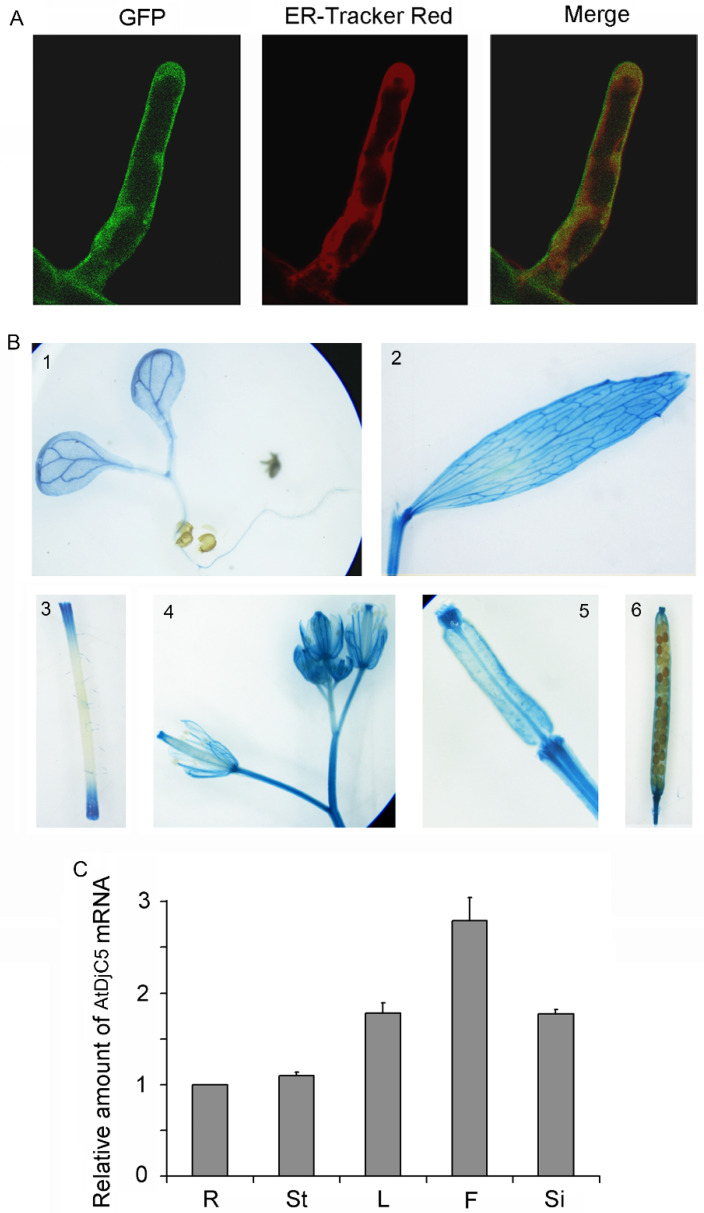
Spatiotemporal expression pattern and subcellular localization of AtDjC5 in *Arabidopsis thaliana*. (**A**) Analysis of AtDjC5 subcellular localization in the root hair cells. Transgenic *Arabidopsis* seedlings harboring *P35S::AtDjC5-sGFP* were stained with ER-Tracker Red. GFP (green) and ER-Tracker Red (red) fluorescence in the root hair cells was observed using a confocal laser microscope. In the photomicrographs on the right, a merge of the two images is presented, and yellow fluorescence was visible in the cells. (**B**) Transgenic plants harboring *PAtDjC5::GUS* were histochemically stained. GUS staining in 1, 6-day-old seedling; 2, cauline leaf; 3, stem; 4, anthotaxy; 5, young silique; 6, mature silique. (**C**) Tissue-specific expression of the *AtDjC5* gene by Q-PCR. The expression level in roots was set to 1 and used for normalization. Data are the means ± SDs of three biological replicates. R, roots; St, stems; L, leaves; F, flowers; Si, siliques.

**Figure 5 ijms-23-13134-f005:**
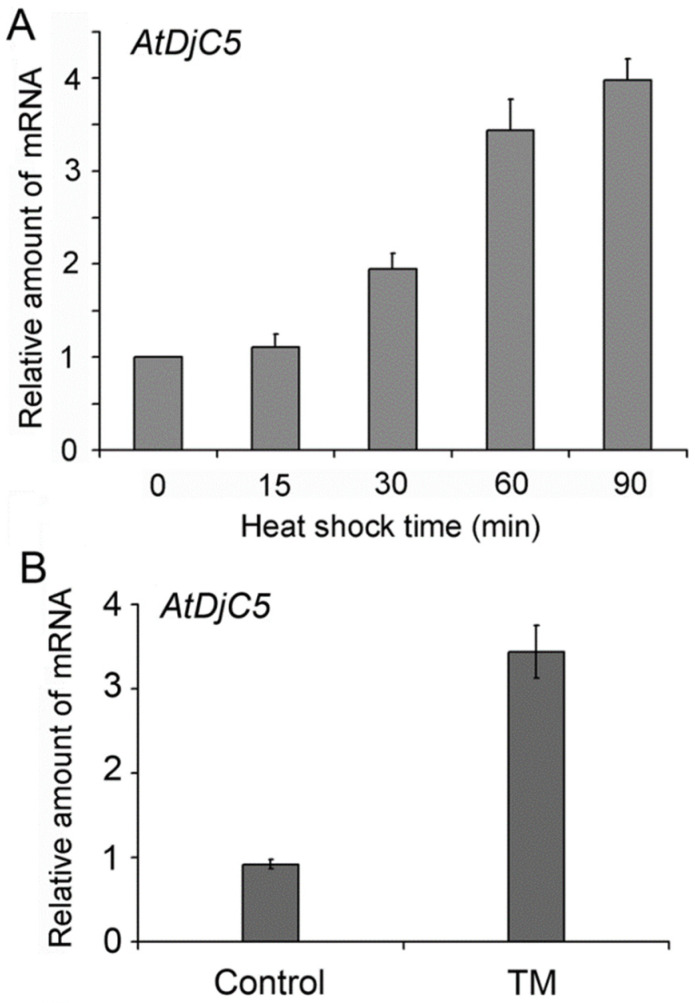
The effects of heat or TM treatment on the expression of the *AtDjC5* gene by Q-PCR. (**A**) Time course of heat-induced *AtDjC5* expression. Ten-day-old seedlings grown at 22 °C were heated at 37 °C for the length of time indicated. The level of *AtDjC5* in the unheated seedlings was normalized to 1. (**B**) The effect of TM treatment on *AtDjC5* expression. Ten-day-old seedlings grown at 22 °C were treated with 5 μg/mL TM for 2 h. The level of *AtDjC5* in the nontreated seedlings was normalized to 1. Q-PCR was performed with *AtDjC5*-specific primers (Appendix A). *Actin* was used as an internal control. Data are the mean ± SD from three independent experiments.

**Figure 6 ijms-23-13134-f006:**
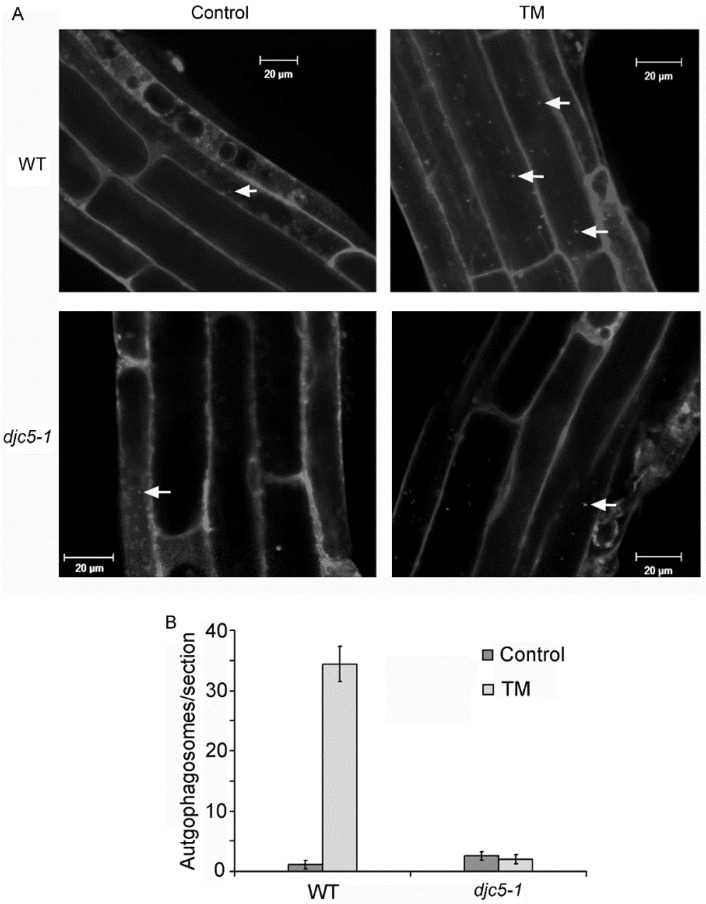
*AtDjC5* knockout inhibited TM-induced autophagy. WT, wild-type plants; *djc5-1*, a T-DNA insertion mutant line for the *AtDjC5* gene; Control, nontreated roots; TM, roots treated with 5 μg/mL TM for 8 h. (**A**) MDC-stained roots before and after TM treatment were observed using a confocal microscope. Arrows indicate autophagosomes. Bar = 20 μm. (**B**) The numbers of MDC-stained autophagosomes per root section before and after TM treatment were counted, and the average numbers were determined for 20 seedlings per treatment. Error bars represent SD.

**Figure 7 ijms-23-13134-f007:**
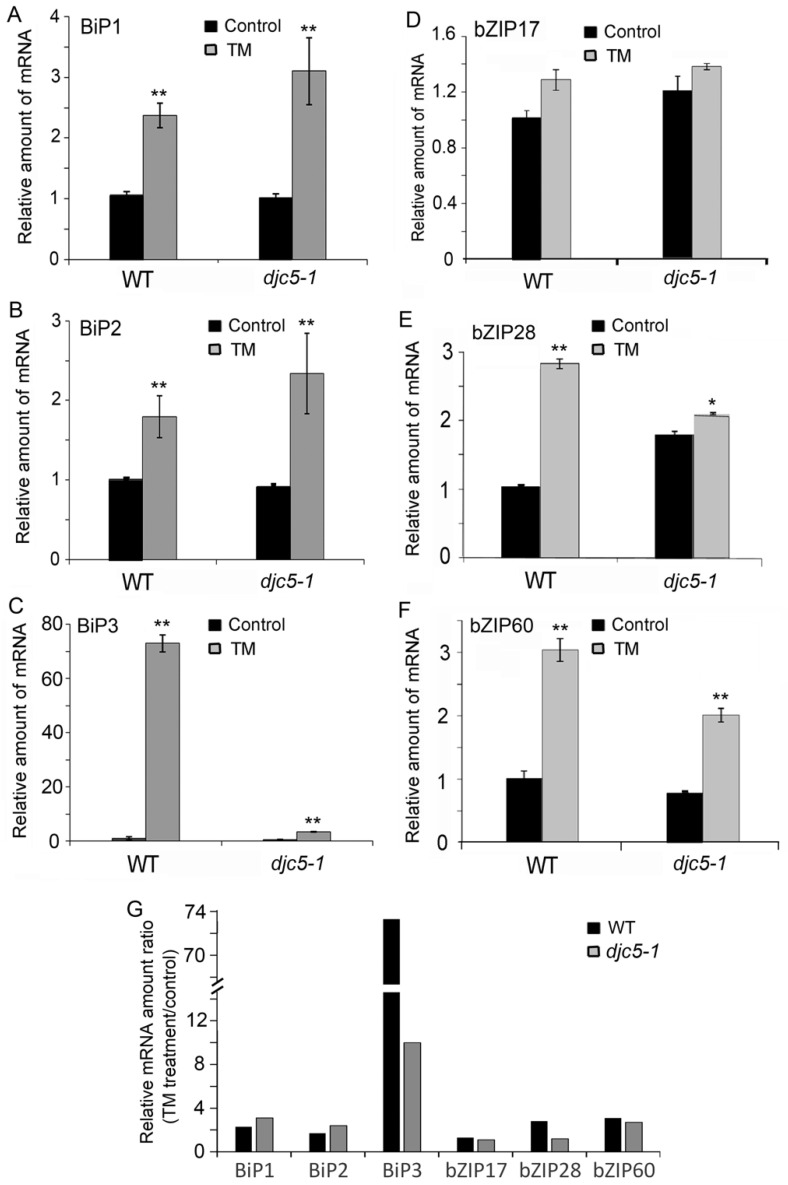
The effects of *AtDjC5* knockout on the expression levels of the three BiP and three bZIP genes before and after TM treatment. WT, wild-type plants; *djc5-1*, a T-DNA insertion mutant line for the *AtDjC5* gene; Control, nontreated seedlings; TM, seedlings treated with 5 μg/mL TM for 30 min. (**A**–**F**) Analysis of the expression levels of the *BiP1*, *BiP2*, *BiP3*, *bZIP17*, *bZIP28*, and *bZIP60* genes in WT and *djc5-1*. Q-PCR was performed with gene-specific primers (Appendix A). *Actin* was used as an internal control. The expression levels of these genes in the nontreated WT seedlings were set to 1 and used for normalization. Data are the mean ± SD from three independent experiments. Asterisks indicate significant differences from the control group. (**G**) Comparison of the relative mRNA amount ratios (TM treatment/control) of *BiP1*, *BiP2*, *BiP3*, *bZIP17*, *bZIP28*, or *bZIP60* between WT and the *djc5-1* seedlings. (* *p* < 0.05, ** *p* < 0.01).

## Data Availability

Data is contained within the article and Appendix A.

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
