# Peer review of "The Arabidopsis J-Protein AtDjC5 Facilitates Thermotolerance Likely by Aiding in the ER Stress Response"

_ijms, 2022, doi:10.3390/ijms232113134_

Round 1
Reviewer 1 Report
*In the abstract better say tunicamycin than TM.
*Reading the introduction, I would have liked to know why the authors decided to study AtDjC5 and not some of the other 120 J-proteins identified in the Arabidopsis genome.
* Why the artificial miRNA lines djc5-2 and djc5-3 were not included together the T-DNA insertion mutant djc5-1 in the results 3.1 to study also its basal thermotolerance too? would have greatly increased the credibility of such results. I suggest the authors to include those data.
*In the transgenic lines that overexpress AtDjC5 OE1 and OE2 its expression levels increase more than 100 times Fig 3A, this level of expression is necessary to achieve a significant increase in survival as show in Fig 3C? Have you studied lines with less expression? Is the expression level of AtDjC5 related to heat shock survival rate? If you have these data, it would be good to include and discuss them, at least as a supplement.
* Results 3.4, why the induced autophagy and the ER stress response was done only with djc5-1 seedlings and not also with the artificial miRNA lines djc5-2 and djc5-3?
* The discussion is somewhat short, half of it is dedicated to remembering part of the introduction, this part could be omitted. I Missed discussing what is special about the protein AtDjC5, compared to the other 120 J-proteins, and more importantly, compared AtDjC5 to the other Arabidopsis proteins described with implication in thermotolerance. it would be clarifying to include a figure that clearly shows the differences between t-proteins, such as a phylogenetic tree or a figure that compares the main protein domains of them. I would like to know of the J-proteins which are the most similar to AtDjC5, what is known about them, it would be expected that they would respond the same??
*In the Lines 216, 226, 328, 354, 396 atj5-1, I think must be changed by djc5-1
Reviewer 2 Report
The work of Shen et al. is well written with well-designed figures. It focus on heat stress although some results (figures 7 and 8), go further into ER stress triggered by tunicamicyn (TM) without connecting concepts like autophagy with heat stress. Besides this two figures, the paper stand by its own merits except that figure 4A is the only evidence that authors provide to sustain the ER localization of the product of gene At5g03030. Nevertheless, a previous report indicated that this protein is localized in the mitochondria and termed this gene PAM18-3 (https://link.springer.com/article/10.1007/s11103-020-01095-8).
Due the poor quality of figure 4A present on this manuscript, I must request that authors provide a better image plus additional evidence that sustain the localization of this protein on the ER. Otherwise, the concept and information behind this report must be revised to avoid the disemination of incomplete data related to the role of a mitochondria localized protein on heat and ER stress response.
Round 2
Reviewer 1 Report
I think the authors have responded favorably to all my suggestions. I accept the current version.
Author Response
Thank you for your approval. There is nothing to response.
Reviewer 2 Report
Please correct this section (MitoTracker was not used on this study as far as authors report):
2.9. Subcellular localization of AtDjC5 174
Roots of 5-day-old transgenic seedlings harbouring P35S::AtDjC5-sGFP were incu-175 bated in 1 μM ER-Tracker Red (Molecular Probes, Carlsbad, CA) solution for 30 min in 176 the dark. Green fluorescent protein (GFP) fluorescence and MitoTracker Red fluorescence 177 were observed using a laser scanning confocal microscope (FV3000; OLYMPUS, Tokyo 178 Prefecture, Japan) with excitation/emission wavelengths of 488/510 nm for GFP and 179 587/615 nm for ER-Tracker Red.
Author Response
Thank you for your kindly suggestion. We have modified this as you say. Please see the revised2 MS.